# Ultrasound-Guided Hydrodissection with Needle Stabilization: An Innovative Nerve-Sparing Approach to Remove a Contraceptive Implant Causing Ulnar Neuropathy

**DOI:** 10.3390/diagnostics15162106

**Published:** 2025-08-21

**Authors:** Yeui-Seok Seo, HoWon Lee, Jihyo Hwang, Chanwool Park, MinJae Lee, Yonghyun Yoon, HyeMi Yu, Jaeik Choi, Gyungseog Ko, Daniel Chiung-Jui Su, Keneath Dean Reeves, Teinny Suryadi, Anwar Suhaimi, King Hei Stanley Lam

**Affiliations:** 1Department of Anatomy, Catholic Institute for Applied Anatomy, College of Medicine, Soeul 06591, Republic of Korea; prssys@area88ps.com; 2Department of Orthopedic Surgery, Hallym University Gangnam Sacred Heart Hospital, 1 Singil-ro, Yeongdeungpo-gu, Seoul 07441, Republic of Korea; lhwghm@gmail.com (H.L.); hwangjihyo36@gmail.com (J.H.); 3IncheonTerminalOrthopedics, Inha-ro 489beon-gil, Namdong-gu, Incheon 21574, Republic of Korea; humanpcw94@gmail.com (C.P.); mjlee951224@gmail.com (M.L.); 4International Association of Regenerative Medicine, Namdong-gu, Incheon 21574, Republic of Korea; 5MSKUS, 1035 E. Vista Way #128, Vista, CA 92084, USA; 6The Board of Clinical Research, The International Association of Musculoskeletal Medicine, Kowloon, Hong Kong; dr.daniel@gmail.com (D.C.-J.S.); painfreedoc22@gmail.com (T.S.); anwar@ummc.edu.my (A.S.); 7Department of Plastic Surgery, BIO Plastic Surgery Clinic, Seoul 06035, Republic of Korea; myangel315@naver.com; 8Choi’s Rehabilitation Clinic, 159, Gwangdeok 1-ro, Danwon-gu, Ansan-si 15476, Republic of Korea; jic121@hanmail.net; 9Ko GyungSeog Orthopaedic Clinic, 551, Ogeum-ro, Songpa-gu, Seoul 05764, Republic of Korea; kgs8368@hanmail.net; 10Department of Physical Medicine and Rehabilitation, Chi Mei Medical Center, Tainan 710, Taiwan; 11A Tempo Regeneration Center for Musicians, Tainan 700, Taiwan; 12Independent Researcher, Roeland Park, KS 66205, USA; deanreevesmd@gmail.com; 13Department of Physical Medicine and Rehabilitation, Hermina Podomoro Hospital, North Jakarta 14350, Indonesia; 14Department of Physical Medicine and Rehabilitation, Medistra Hospital, South Jakarta 17431, Indonesia; 15Physical Medicine and Rehabilitation, Synergy Clinic, West Jakarta 11520, Indonesia; 16Department of Rehabilitation Medicine, Universiti Malaya, Kuala Lumpur 50603, Malaysia; 17Faculty of Medicine, The University of Hong Kong, Hong Kong; 18Faculty of Medicine, The Chinese University of Hong Kong, New Territory, Hong Kong; 19The Board of Clinical Research, The Hong Kong Institute of Musculoskeletal Medicine, Kowloon, Hong Kong

**Keywords:** ulnar neuropathy, contraceptive agents, female, peripheral nerve injuries, minimally invasive surgical procedures, ultrasonography, interventional, hydrodissection, foreign body removal, 5% dextrose, without local anesthetics

## Abstract

**Background and Clinical Significance:** Non-palpable migrated contraceptive implants pose significant challenges for removal and are associated with neurovascular complications. Traditional open surgery near nerves is associated with postoperative morbidity. Migrated or deeply embedded implants near critical structures can result in severe complications, such as neuropathy, and their removal typically requires open surgical intervention. **Case Presentation:** We report a novel, minimally invasive, ultrasound (US)-guided technique for removing a migrated etonogestrel Implanon^®^ implant that caused ulnar neuropathy. A 38-year-old woman presented with severe neuropathic pain and paresthesia (NPRS 10/10; QuickDASH 55) along her left ulnar nerve following multiple failed removal attempts that induced deep migration. US confirmed the proximity of the implant to the ulnar nerve. Initial US-guided removal exacerbated her symptoms. Hydrodissection (HD) with 50 mL of 5% dextrose in water (D5W) without local anesthetic (LA) was performed to reduce inflammation and achieve separation. The implant migrated proximally during extraction. An additional HD with 50 mL of D5W without LA distally repositioned the implant. Percutaneous stabilization using a 25-gauge needle enabled secure removal. The intact 4 cm implant was extracted under real-time US guidance without open surgery. The patient experienced immediate symptom relief (NPRS 2/10; QuickDASH 4.5 at one month) and full resolution (NPRS 0/10; QuickDASH 0) with no motor deficits at one year. **Conclusions:** This case represents the first documented percutaneous removal of a nerve-adherent implant using combined US-guided D5W HD and needle stabilization, marking a paradigm shift in the management of such cases. This approach confirms the safety of US-guided foreign body removal using HD for nerve-adjacent implants and demonstrates the efficacy of combining D5W HD with needle stabilization. Surgical morbidity was avoided, while excellent long-term outcomes were achieved.

## 1. Introduction

Musculoskeletal ultrasound (US) has become an indispensable tool for guiding interventional procedures due to its real-time visualization capacity, lack of ionizing radiation, and cost-effectiveness [1]. In addition to these advantages, US enables continuous monitoring of soft tissues and needle trajectories, facilitating accurate targeting while minimizing neurovascular risk [1]. The clinical applications of US guidance have grown substantially and encompass joint injections, nerve blocks, and specialized percutaneous tendon procedures. Notably, in cases of calcific shoulder tendinitis, US-guided barbotage yields good to excellent results in over 90% of patients, with fewer than 1% requiring surgery, underscoring the high success rates and safety profiles of US-guided treatments [2].

In the context of foreign body and implant management, particularly for contraceptive implants, US offers distinct advantages. Its sensitivity allows the detection of non-palpable implants, which typically appear as echogenic rods with characteristic reverberation shadows, facilitating accurate localization and removal. Studies have demonstrated that impalpable Implanon^®^ rods can be successfully located and extracted using US guidance in nearly all cases [3]. Real-time guidance is particularly valuable when implants migrate or are positioned near critical structures, as blind removal attempts pose substantial risks.

While contraceptive arm implants are generally inserted subdermally and removed through palpation, approximately 3–5% of cases present challenges when the device becomes deeply placed, fractured, or migrated, complicating retrieval [3,4,5,6,7,8]. These complex scenarios can lead to serious complications, including reported cases of implants impinging on neurovascular structures, such as the ulnar and median nerves, resulting in neuropathy [4,6,7,8,9]. In these high-risk situations, meticulous imaging and specialized care are essential to prevent permanent neurological injuries [7,8,9]. We present a unique case of a 38-year-old woman with neuropathic arm pain caused by a retained Implanon implant directly abutting the ulnar nerve. The implant was successfully removed using US-guided hydrodissection (HD) and percutaneous needle stabilization techniques. This case highlights the value of high-resolution US in managing non-palpable implants while avoiding nerve damage and contributes to the evolving literature on minimally invasive solutions for complex implant removals.

Hydrodissection (HD) is an advanced ultrasound-guided technique that involves injecting fluid to separate nerves from compressive structures [10], such as fascia [11,12], scar tissue [13], and implants. Unlike conventional ultrasound guidance, which primarily visualizes anatomy, HD creates a dynamic, protective fluid plane between critical structures. This technique represents a paradigm shift in high-risk foreign body removal by addressing a key limitation of traditional approaches—namely, the inability to shield nerves during instrument manipulation near neural structures—thereby reducing the risk of iatrogenic injury.

When 5% dextrose in water (D5W) is used as the injectate without local anesthetics, HD provides three protective effects. First, it mechanically separates tissue planes to isolate nerves from implants [10]. Second, it modulates inflammation by downregulating TRPV1 [14,15] channels and mitigating neurogenic inflammation [16,17,18,19]. Finally, it stabilizes metabolic conditions by correcting perineural glycopenia [17], thereby reducing nerve hyperexcitability. The biocompatibility and iso-osmolality of D5W (277 mOsm/L) ensure the safety of neural structures while enabling real-time feedback from patients. In this pioneering case, we employed ultrasound-guided D5W HD to establish a protective buffer between the ulnar nerve and a contraceptive implant, achieving the first documented percutaneous extraction in which HD successfully separated a nerve-adherent implant without the need for open surgery. Additionally, percutaneous needle stabilization was used to manage the implant’s mobility and slipperiness, a previously undocumented approach that offers an alternative to open exploration in such cases.

## 2. Case Presentation

A 38-year-old Asian woman presented with a three-day history of progressively worsening paroxysmal pain characterized by burning, shooting, and electric shock-like sensations, accompanied by persistent tingling, numbness, and paresthesia radiating from the medial left elbow and forearm to the medial hand. Her medical history included the placement of a contraceptive etonogestrel implant (Implanon^®^) in the left upper arm, which had been asymptomatic for three years, coinciding with the end of its intended lifespan. Multiple removal attempts at a local clinic were unsuccessful due to the significant depth of the implant. Each manipulation exacerbated neuropathic symptoms along the ulnar nerve distribution, leading to increasingly severe paresthesia, which precluded further attempts. Despite the efforts of three obstetricians over three consecutive days, removal remained unsuccessful (Figure 1). The patient was discharged with an urgent referral to our orthopedic center three days after the final attempt, without prior radiological investigation.

Within 48–72 h of the final unsuccessful removal attempt, the patient developed new-onset localized tingling and pain at the medial elbow, which was exacerbated by arm movement. In the nights preceding her initial evaluation, her symptoms intensified dramatically, resulting in intolerable nocturnal paresthesia characterized by severe “pins and needles” sensations that disrupted her sleep quality. As a nail salon proprietor, she became unable to perform occupational or daily activities due to the debilitating symptoms. Her sleep was further impaired by progressive night-time pain. Her medical history was otherwise unremarkable, and there was no documented trauma to account for her presentation.

Physical examination revealed three small wounds on the medial left arm from prior removal attempts (Figure 1). The implant was not palpable. Tinel’s sign was positive in the medial arm, eliciting radiating pain along the ulnar nerve distribution. Sensory testing revealed decreased sensation in the medial antebrachial cutaneous and ulnar nerve territories (Figure 2). Although the patient exhibited guarded arm movements due to pain, a comprehensive neurological assessment revealed no motor deficits. Her pain was rated at maximum severity (numeric pain rating scale (NPRS) score: 10/10), significantly impairing her function, as reflected by a Quick Disabilities of the Arm, Shoulder, and Hand (QuickDASH) score of 55, indicating a marked upper extremity dysfunction. Clinical evaluation strongly suggested deep implant migration with probable neurovascular impingement.

### 2.1. Prior Imaging Included Radiography (Figure 3), Which Confirmed the Implant’s Location, and Ultrasound (Figure 4), Which Demonstrated Its Proximity to the Ulnar Nerve

Plain radiography of the left arm revealed no bony deformities or structural abnormalities that could account for the patient’s neurological symptoms. However, imaging revealed a linear radiopaque structure consistent with the residual Implanon device in the medial aspect of the arm (Figure 3).

**Figure 3 diagnostics-15-02106-f003:**
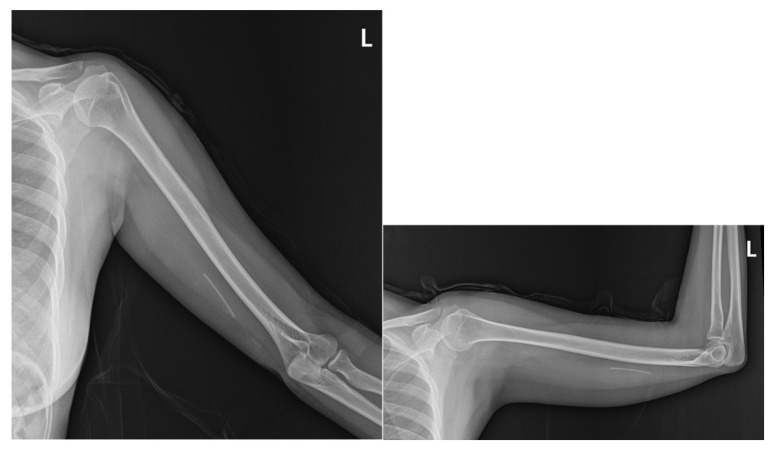
Plain radiograph of the left arm showing a linear radiopaque structure consistent with the retained Implanon^®^ device. No bony deformities or structural abnormalities that could account for the patient’s neurological symptoms were noted.

Radiographic confirmation prompted further evaluation with US imaging of the left medial arm due to clinical suspicion of implant-related nerve irritation. Sonographic assessment using a high-frequency linear transducer (GE Linear L4-20t-RS; General Electric, Boston, MA, USA) revealed a linear echogenic structure with a reverberation artifact just above the medial humerus, approximately 5 cm proximal to the medial epicondyle.

**Figure 4 diagnostics-15-02106-f004:**
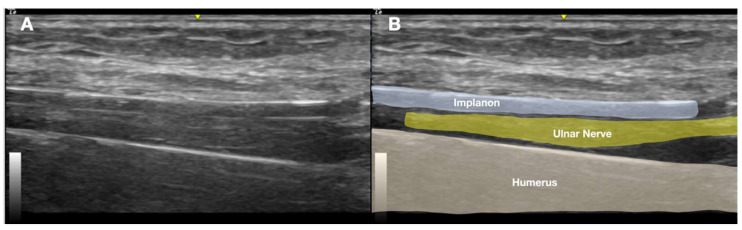
Long-axis ultrasound image of the medial arm demonstrating the Implanon^®^ device positioned deep in the muscle layer adjacent to the ulnar nerve within the neurovascular bundle. (**A**) Ultrasound image. (**B**) Labeled explanation.

US findings localized the echogenic structure beneath the muscle layer, adjacent to the ulnar nerve, and within the neurovascular bundle of the upper arm (Figure 4 and Figure 5A). No neuroma or nerve enlargement was observed; however, dynamic US demonstrated apparent direct contact between the implant and nerve sheath. The clinical impression was of a migrated, deeply seated, and unstable implant in close proximity to the ulnar nerve. Intermittent nerve irritation or impingement, likely exacerbated by daily movement and previous failed removal attempts, was considered the probable cause of neuropathic pain. After obtaining informed consent, the decision was made to proceed with minimally invasive US-guided removal as an alternative to open surgical exploration.

### 2.2. The Procedure

The procedure was performed on the first day of the patient’s visit to our center under local anesthesia in an outpatient intervention suite. With the patient positioned supine and the left arm abducted, the team re-explored the previous incision site under sterile conditions. US confirmed the retained Implanon 2 cm distal to one of the incision sites and immediately adjacent to the ulnar nerve. Initial direct US-guided removal attempts were unsuccessful due to significant nerve irritation triggered by manipulation. To address this, approximately 50 mL of 5% dextrose in water (D5W) without local anesthesia (LA) was injected via US-guided HD at the distal aspect of the implant. This approach aimed to achieve both a biochemical anti-inflammatory effect and mechanical separation between the implant and adjacent nerve (Figure 6). The HD technique successfully creates a protective fluid buffer around neural structures.

Despite the ulnar nerve protection achieved with D5W HD, the implant could not be secured due to its low-friction surface, and it evaded capture with mosquito forceps under US guidance. Subsequent fluoroscopic correlation using C-arm imaging confirmed iatrogenic proximal migration of the implant toward the axillary region. Under combined US–fluoroscopic guidance, the implant was re-identified in the proximal upper arm. A second targeted HD session was performed under US guidance, during which an additional 50 mL of D5W without LA was injected from proximal to distal along the anticipated tract to facilitate distal repositioning of the implant back to the distal arm. To overcome the challenge of continued implant mobility, the proximal tip of the implant was percutaneously penetrated and stabilized using a 25-gauge (G) needle under US guidance (Figure 5B). This innovative stabilization technique enabled secure grasping of the implant with mosquito forceps and successful extraction through the previous incision under real-time US visualization (Figure 5C).

The entire 4 cm Implanon rod segment was removed intact (Figure 7). No additional incisions or surgical dissection were required. Hemostasis was achieved via manual compression, and the wound was closed with adhesive strips. The patient tolerated the procedure well, with no immediate complications. Complete removal was confirmed using concurrent C-arm fluoroscopy and US imaging (Figure 8 and Figure 9). Notably, the patient reported immediate and substantial symptom relief, with pain severity decreasing to an NPRS score of 2/10 (80% reduction). The entire procedure—including the initial removal attempt, US-guided HD, the second attempt, further US-guided HD, percutaneous 25G needle stabilization, and final retrieval—was completed within two hours. Post-procedural monitoring for one hour revealed some fluctuation in pain scores. A long-arm splint was applied to maintain elbow extension. Prophylactic antibiotics and one-day NSAID therapy were prescribed due to repeated wound manipulation and local inflammation. The patient was discharged the same day.

### 2.3. Follow-Up and Long-Term Outcomes

The wound was reassessed at our center the day after extraction, with no signs of infection or exudate observed. The patient’s pain level was 1–2/10 on the numeric pain rating scale (NPRS), and she demonstrated full range of motion in the left elbow and wrist.

At follow-up one month post-procedure, the patient exhibited significant clinical improvement. The initial presentation of debilitating radiating forearm pain and tingling had largely resolved, with only minimal residual numbness in the little finger, which continued to improve. Quantitative assessment indicated substantial functional recovery, as evidenced by a decrease in the QuickDASH score to 4.5, reflecting near-complete restoration of upper extremity function. Serial neurological examinations confirmed the absence of motor deficits or new neurological symptoms, supporting the conclusion that the neuropathic pain originated from ulnar nerve irritation secondary to the retained implant.

The long-term outcomes were equally favorable. At one-year follow-up conducted via structured telephone interview, all clinical indicators confirmed sustained therapeutic success. The patient reported an NPRS score of 0, and upper limb function had fully normalized, reflected by a QuickDASH score of 0.

## 3. Discussion

This case illustrates a paradigm shift in the management of migrated contraceptive implants through innovative approaches. First, the observed migration pattern, wherein the implant traveled proximally along the fascial plane or through the epineurium to the axillary region, demonstrates a rare trajectory indicative of a unique pathomechanism of nerve irritation. While the literature documents ectopic implants encasing or penetrating nerve sheaths, such occurrences are extraordinarily uncommon [4,6,7]. Unlike prior reports, such as that of Saeed et al. [20], in which neuropathy resulted from spontaneous migration, our case highlights iatrogenic displacement during unsuccessful removal attempts.

Second, our percutaneous approach—ultrasound-guided hydrodissection (HD) with 5% dextrose in water (D5W) without local anesthetics to separate the nerve from the implant prior to extraction—offers an alternative to conventional open neurolysis, which is typically employed when implants are in contact with or adjacent to neural structures. Instead, this method enables fluid-mediated nerve liberation with real-time visualization [4]. Third, our needle stabilization technique addressed a persistent challenge in US-guided removal: implant mobility during grasping. Percutaneous fixation with a 25G needle enabled secure extraction through existing incisions, representing a technical refinement beyond standard hydrodissection.

While previous studies have described ultrasound-guided removal of deep implants [6,8], our technique advances this approach by integrating 5% dextrose in water (D5W) hydrodissection to separate the nerve from the implant, creating a protective zone for ultrasound-guided manipulation of the implant using surgical instruments and preventing further nerve damage. This method also reduces neurogenic inflammation associated with implant manipulation. Additionally, our technique introduces percutaneous needle stabilization, specifically designed for easily migrating or slippery implants.

To our knowledge, this is the first documented case of successful management of an iatrogenically malpositioned implant using ultrasound-guided hydrodissection with D5W to mitigate mechanical impingement and neurogenic inflammation. The implant was removed despite its slippery nature via ultrasound-guided percutaneous extraction with single-needle stabilization, avoiding the need for additional incisions or open surgery.

The exceptional clinical course, from immediate post-procedural improvement (NPRS 10→2/10) to complete long-term resolution (QuickDASH score of 0 at 1 year), provides compelling evidence that US-guided, minimally invasive removal offers an effective and lasting solution for implant-induced nerve compression.

### Management Strategies for Non-Palpable or Deeply Placed Implants

When managing non-palpable or deeply placed implants, current clinical guidelines recommend imaging localization and specialist referral [3,7,21]. Our experience supports US as the first-line imaging modality, given its lack of ionizing radiation and superior real-time spatial resolution [4,6,7,8]. In cases where US confirms an implant’s precarious location adjacent to neural structures, removal should be undertaken with particular caution.

Although some cases require operative intervention under regional or general anesthesia with wider surgical exposure [4,7], our findings demonstrate that implants in close proximity to neural structures can be removed percutaneously. Traditional surgical management typically involves gaining proximal and distal control of both the implant and nerve via open exploration to prevent traction injury [4,7]. However, open surgical approaches carry risks of postoperative perineural adhesions and undesirable scarring, which may impair functional and cosmetic outcomes. The US-guided HD technique we employed [10] addresses these challenges by creating a protective fluid cushion around the affected nerves.

D5W achieves a glucose concentration approximately 50-fold higher than physiological plasma and tissue fluid levels while maintaining an osmolality (277 mOsm/L) comparable to that of normal saline (308 mOsm/L). Clinical evidence indicates that perineural administration of D5W causes significantly less injection-associated discomfort than sterile water [22]. Additionally, studies have confirmed the absence of neurotoxic effects on neural tissues following D5W exposure [22,23,24]. To date, no complications attributable to dextrose-based hydrodissection have been reported [25,26].

The application of D5W provides immediate mechanical separation between the implant and ulnar nerve through controlled fluid dissection. Unlike local anesthetics, D5W preserves sensory and motor functions by avoiding neural blockade, thereby allowing real-time patient feedback during instrument manipulation and reducing the risk of iatrogenic nerve injury.

D5W has therapeutic advantages over normal saline in managing neuropathy and neurogenic inflammation. Research indicates that D5W not only provides effective mechanical separation but also exerts beneficial metabolic effects that enhance neural recovery and reduce inflammation more significantly than normal saline [26,27,28]. This is particularly important in conditions characterized by neuropathic pain, as D5W may improve neuronal cell function [16,27]. Furthermore, D5W exhibits multimodal anti-inflammatory effects on compressed neural tissues through distinct biochemical pathways. Specifically, glucose-mediated modulation attenuates transient receptor potential vanilloid receptor-1 (TRPV1) in sensory neurons [27,29,30] and reduces neurogenic inflammation [16,17,26], contributing to improved outcomes in patients undergoing nerve-adherent implant removal.

The clinical safety and efficacy profiles of D5W further support its utility. The absence of local anesthetics mitigates several associated risks, including temporary sensory and motor blockade, which may obscure nerve traction injuries, chondrotoxicity, and allergic reactions. Additionally, avoiding corticosteroids [30,31] eliminates concerns regarding tendon weakening, glycemic dysregulation, and delayed tissue healing.

Moreover, D5W HD enables pain-tolerant extraction while maintaining protective sensation. The immediate post-procedural pain reduction (NPRS 10→2/10) without sensory blockade indicates that symptom resolution was primarily attributable to implant removal rather than anesthetic effects. The absence of neuropathic sequelae at the 1-year follow-up, along with complete functional recovery (QuickDASH score of 0), further indicates the potential therapeutic advantages of this approach over solutions containing anesthetics.

Ensuring complete implant removal is critical, particularly when a fracture is suspected. Our protocol emphasizes meticulous US inspection post-extraction to verify that no residual fragments remain [6]. In this case, verification confirmed the retrieval of an intact 4 cm rod.

While Implanon^®^ removal is conventionally performed by obstetrician–gynecologists, cases presenting with neurological symptoms necessitate referral to clinicians proficient in ultrasonography-guided extraction. Operator expertise in ultrasound-guided interventions—particularly nerve hydrodissection and needle placement—is critical, as inadequate visualization during removal can lead to iatrogenic neural injury [4]. Clinicians employing these advanced techniques must possess specialized ultrasound and procedural training.

Accumulating evidence supports the use of US guidance as the preferred method for locating and removing non-palpable contraceptive implants. Early evidence from case series in family planning clinics, such as that by Singh et al. [3], demonstrated successful identification and removal of 21 “lost” Implanon devices under US guidance [3]. Subsequent studies, including those by Patel et al. [32] and Persaud et al. [33], reinforced these findings, achieving high success rates with ultrasound-guided removal [32,33]. Recent advancements have further refined percutaneous removal techniques [34], establishing US-guided localization and removal as a reliable first-line approach for challenging implants, with success rates and safety profiles comparable to those of other established US-guided procedures [2].

A comparison of the surgical techniques for implant removal is summarized in Table 1. Metrics include operative time, incision size, risk of nerve injury, soft tissue damage, potential for conversion to open surgery, learning curve, and complications associated with each method.

Several limitations should be considered when interpreting the findings of this case report. As a single-patient experience, it does not establish the comparative efficacy of the described technique. While the successful outcome suggests the clinical utility of US-guided HD and removal, more extensive studies or case series are needed to improve generalizability. The absence of pre-and post-procedure nerve conduction studies limits rigorous assessment of neurological recovery; however, the documented sensory deficits in the medial antebrachial cutaneous and ulnar nerve distributions, which completely resolved at both one-month and one-year follow-up, may serve as surrogate indicators of neural functional restoration.

The specialized skill set required for this procedure, which should be performed by an orthopedic surgeon with extensive US experience, may not be widely available, potentially limiting its broader adoption. This highlights the need for comprehensive training programs and clear referral pathways, especially for primary care providers and gynecologists who frequently encounter such cases [21].

Technical risks associated with US-guided HD also warrant consideration. Iatrogenic nerve injury is a significant concern, particularly when the procedure is performed by less experienced operators who may inadvertently damage adjacent neurovascular structures. The success of the procedure relies on operator expertise in US interpretation and interventional techniques, contributing to variability in outcomes across clinical settings. Although US provides excellent real-time visualization, its limitations in detecting deeply located implants or complex anatomical structures may increase the risk of incomplete removal.

The generalizability of our findings is limited by the single-case nature of this report, as variations in patient anatomy and clinical presentation may substantially influence outcomes. While follow-up demonstrated excellent short- to mid-term results, it did not address potential long-term complications or recurrences. Standard procedural risks, including infection and inflammatory reactions to injectables, must also be considered in clinical decision-making. These limitations highlight the importance of appropriate patient selection, specialized operator training, and structured postoperative monitoring for optimizing outcomes.

## 4. Conclusions

This case report supports US-guided percutaneous removal as a first-line strategy for managing nerve-compromising implant complications, offering distinct advantages over traditional approaches. Compared with blind methods or fluoroscopic guidance, US provides superior soft tissue resolution, enhancing nerve preservation. Unlike open surgical exploration, this minimally invasive approach reduces morbidity while maintaining efficacy, as evidenced by the patient’s rapid recovery and sustained resolution of symptoms.

Three practical implications emerge from this case:Early imaging evaluation: Prompt imaging is essential for non-palpable implants associated with neurological symptoms.Specialized technical skills: Proficiency in needle stabilization and HD techniques is crucial for achieving optimal outcomes.Centralized referral pathways: Referral systems should be established for complex cases.

As global contraceptive implant use increases, this case illustrates how advanced US techniques can reshape the management of rare but serious complications. Future priorities include standardizing HD protocols, developing simulation-based training for needle stabilization, and establishing multicenter registries to monitor outcomes. Such initiatives will help ensure that these minimally invasive solutions reach their full potential in enhancing patient safety worldwide.

Future research should prioritize refining minimally invasive techniques, optimizing HD solutions and instrument design, and aggregating outcome data through multicenter registries. Given the rarity of implant-related neuropathies, collaborative efforts are essential for developing evidence-based management protocols. Preventive measures, such as correct insertion techniques to ensure true subdermal placement and immediate post-insertion palpation verification, are equally important. A standardized approach incorporating prompt imaging for non-palpable implants and early involvement of experienced removal teams should be adopted as the clinical standard to minimize patient morbidity.

### Key Lessons

**Prevention potential**: Proper insertion technique and post-placement palpation can prevent most migration-related complications [21].**Migration matters**: Iatrogenic implant migration (as opposed to spontaneous migration) can cause severe neuropathy, necessitating different management strategies than those traditionally described [4,20].**Imaging imperative**: US must be the first-line approach for non-palpable implants with neurological signs, as blind removal attempts pose a risk of nerve injury [4,21].**Technique triad**: The combination of (a) 5% dextrose HD, (b) 25G needle stabilization, and (c) real-time US visualization enables successful percutaneous removal of nerve-adherent implants.**Training gap**: Despite optimal techniques, operator expertise remains a limiting factor, highlighting the urgent need for specialized training programs [4,21].**Research priorities**: Multicenter registries should track (a) optimal HD volumes, (b) long-term nerve outcomes, and (c) cost–benefit analyses compared to surgery [6].

## Figures and Tables

**Figure 1 diagnostics-15-02106-f001:**
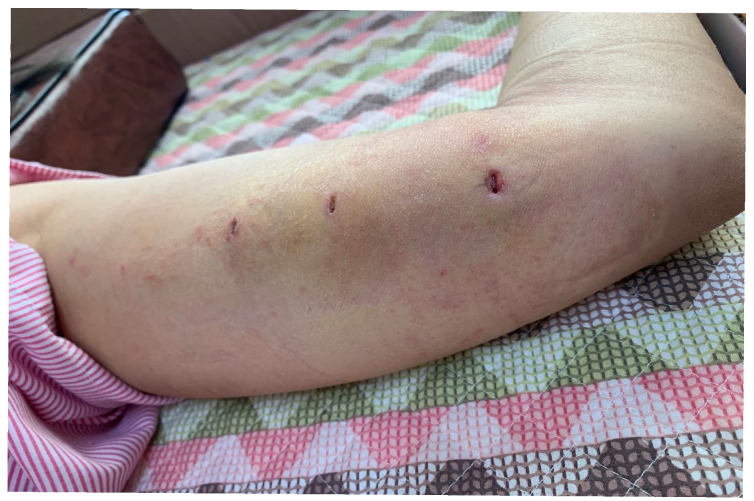
Multiple wounds observed on the inner side of the left arm.

**Figure 2 diagnostics-15-02106-f002:**
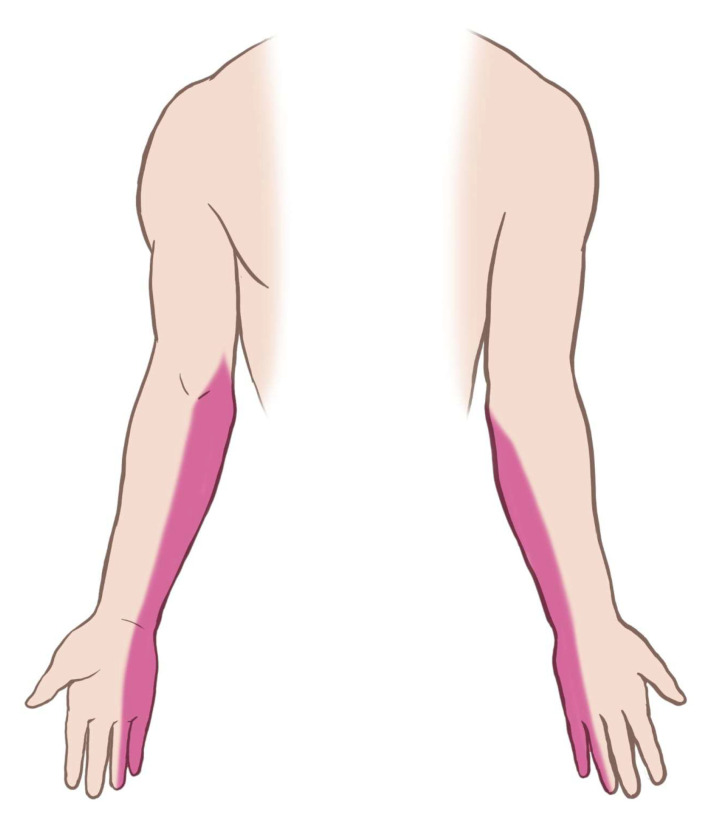
Illustration depicting patient-reported areas of paresthesia and sensory deficits, primarily affecting the medial forearm and hand.

**Figure 5 diagnostics-15-02106-f005:**
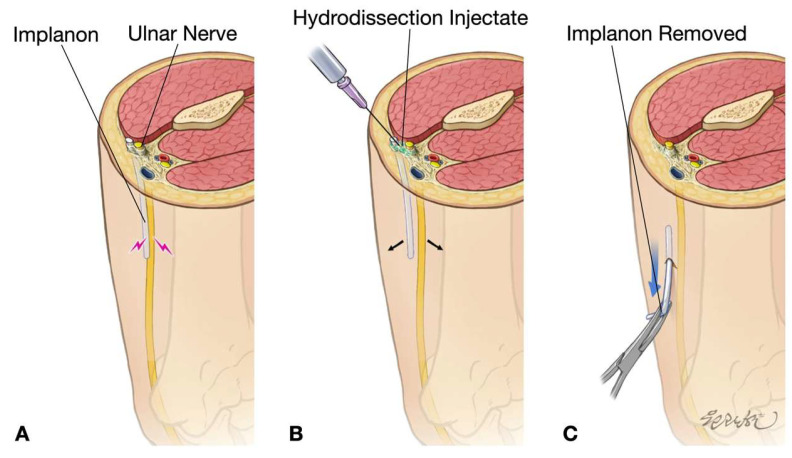
(**A**) Close relationship between Implanon^®^ and the ulnar nerve. (**B**) Stabilization of one end of the Implanon device using a 25G needle. (**C**) Removal of Implanon.

**Figure 6 diagnostics-15-02106-f006:**
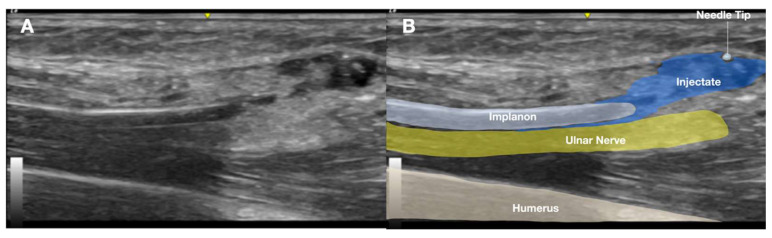
Ultrasound-guided hydrodissection of the distal aspect of the Implanon^®^ device using dextrose 5% in water (D5W) without local anesthetic. The injectate creates a hypoechoic fluid plane between the implant and the adjacent ulnar nerve, facilitating both mechanical separation and biochemical relief of perineural inflammation. (**A**), ultrasound image; (**B**), labeled explanation.

**Figure 7 diagnostics-15-02106-f007:**
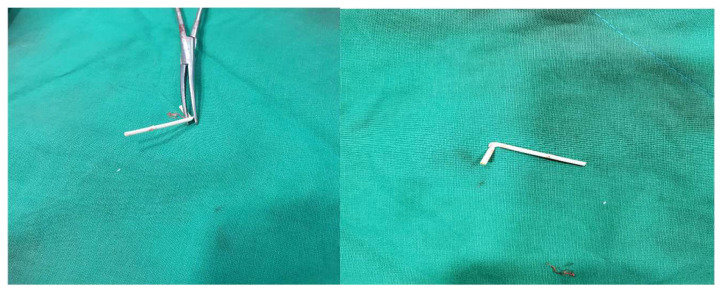
Gross photograph of the fully removed Implanon^®^ rod, approximately 4 cm in length. The device was extracted intact without fragmentation, confirming complete retrieval and eliminating the risk of residual implant material.

**Figure 8 diagnostics-15-02106-f008:**
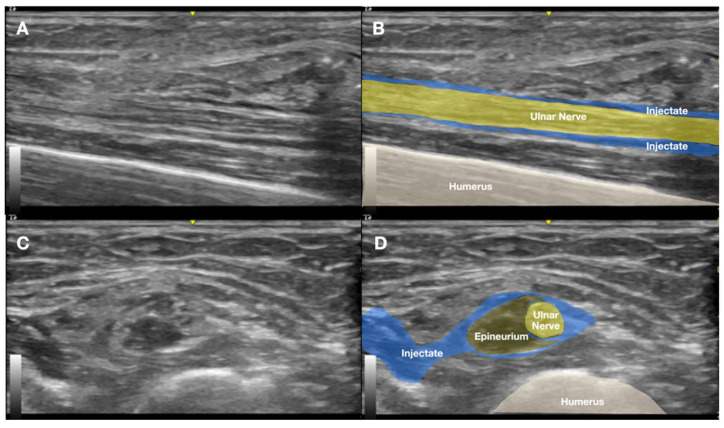
Post-procedural ultrasound image of the medial arm confirming complete removal of the Implanon^®^ device. No residual foreign material was visible. Swelling of the posteromedial ulnar nerve segment was noted without surrounding implant remnants, indicating perineural sheath irritation. (**A**) Long-axis view of the ulnar nerve. (**B**) Corresponding labeled explanations. (**C**) Short-axis view of the ulnar nerve. (**D**) Corresponding labeled explanations.

**Figure 9 diagnostics-15-02106-f009:**
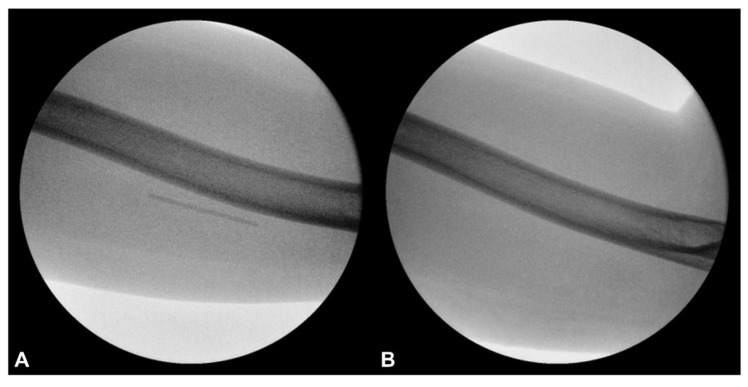
C-arm fluoroscopy images demonstrating the position of the Implanon^®^ device before and after removal. (**A**) Pre-procedural image showing a linear radiopaque structure consistent with the retained implant in the medial arm. (**B**) Post-procedural image confirming complete removal, with no residual radiopaque material visible at the previously identified location.

**Table 1 diagnostics-15-02106-t001:** Comparative analysis of deeply located implant removal techniques.

Characteristic	Open Removal	US-Guided Removal	US-Guided HD + Stabilization + Removal
Target Indications	Excessively difficult implants	Deep implants *(no mention of proximity to nearby nerves in the literature)*	Nerve-adjacent or impinging implants
Operation Time	Long (>2 h)	Short (<30 min)	Moderate (30 to 60 min)
Scar Size	Large incision	Small incision	Small incision
Nerve Injury Risk	Possible	Possible	Minimal
Soft Tissue Damage	Extensive	Moderate	Minimal
Migration Risk	Negligible	Possible	Reduced
Conversion to Open Surgery	N/A	Possible	Rare
Learning Curve	Low	Moderate	High
Complication Rate	Higher	Lower	Minimal

## Data Availability

Data related to this study are included in the manuscript.

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
