# Peer review of "Ultrasound-Guided Hydrodissection with Needle Stabilization: An Innovative Nerve-Sparing Approach to Remove a Contraceptive Implant Causing Ulnar Neuropathy"

_diagnostics, 2025, doi:10.3390/diagnostics15162106_

Round 1

Reviewer 1 Report

Comments and Suggestions for Authors
  1. Title: The phrases before the column should be omitted as it is too long.
  2. Abstract: a) There is no need to provide a percentage here. b) The last sentence should not be numbered.
  3. Keywords: They should be rearranged according to MESH terms.
  4. Introduction: A brief paragraph about hydrodissection should be included.
  5. Case Presentation: a) Does the patient have any previous imaging done before the intervention?
Comments on the Quality of English Language

The text requires thorough language editing for clarity and correctness.

Author Response

We sincerely thank the reviewer for the insightful and constructive comments, which have significantly improved the clarity and scientific rigor of our manuscript. Below, we provide a detailed point-by-point response, indicating how each comment has been addressed in the revised version.

Comment 1 Title: The phrases before the column should be omitted as it is too long.

Response:

Thank you for your suggestion, we have shortened the title:

Ultrasound-Guided Hydrodissection with a Needle Stabilization: An Innovative Nerve-Sparing Approach for Removal of a Contraceptive Implant Causing Ulnar Neuropathy

Comment 2 Abstract:

There is no need to provide a percentage here.

Response:

Thank you for your suggestion. We have removed the percentage “3–5%” from the first sentence of the abstract.

The last sentence should not be numbered.

Response: Thank you for your suggestion.

  • The final sentence has been revised to remove the enumeration. It now reads as a single, cohesive summary of the conclusions without numerical listing.
  • We changed the sentence line 70-73 “The approach confirms the safety of US-guided foreign body removal using HD for nerve-adjacent implants, demonstrates the efficacy of combining D5W HD with needle stabilization, and underscores the ability to avoid surgical morbidity while achieving excellent long-term outcomes.”

Comment 3 Keywords: They should be rearranged according to MESH terms.

Response: Thank you for your suggestion. We have changed the keywords:

Ulnar Neuropathy; Contraceptive Agents, Female; Peripheral Nerve Injuries; Minimal-ly Invasive Surgical Procedures; Ultrasonography, Interventional; Hydrodissection; Foreign Body Removal; 5% dextrose; without local anesthetics.

Comment 4 Introduction: A brief paragraph about hydrodissection should be included.

Response: Thank you for your suggestion, we have added lines 446 to 466:

“Hydrodissection (HD) is an advanced ultrasound-guided technique that involves injecting fluid to separate nerves from compressive structures[10], such as fascia[11, 12], scar tissue[13], or implants. Unlike conventional ultrasound guidance, which primarily visualizes anatomy, HD creates a dynamic protective fluid plane between critical structures. This represents a paradigm shift in high-risk foreign-body removal by addressing a fundamental limitation of traditional approaches: preventing iatrogenic nerve injury during instrument manipulation in proximity to neural structures.

When utilizing 5% dextrose in water (D5W) as the injectate, without local anesthetics, HD achieves three protective effects. First, it mechanically separates tissue planes to isolate nerves from implants [10]. Second, it modulates inflammation by downregulating TRPV1 [14, 15] channels and mitigating neurogenic inflammation [16-19]. Finally, it stabilizes metabolic conditions by correcting perineural glycopenia [17], reducing nerve hyperexcitability. The biocompatibility and iso-osmolality of D5W (277 mOsm/L) ensure the safety of neural structures while facilitating real-time patient feedback. In this pioneering case, we employed ultrasound-guided D5W HD to establish a safety buffer between the ulnar nerve and a contraceptive implant, achieving the first documented percutaneous extraction utilizing HD to separate the nerve from the implant, even when the nerve was adherent to it, without resorting to open surgery. Additionally, the use of percutaneous needle stabilization for implants that are highly slippery and mobile in nature represents another pioneering technique in the literature, as there are currently no effective treatments for such implants apart from open exploration.”

Comment 5. Case Presentation: a) Does the patient have any previous imaging done before the intervention?
Response:
Thank you for your suggestion.

  • Figure 3: X-rays taken prior to the procedure involving the extraction of the Implanon device.
  • Figure 4: Ultrasound imaging obtained prior to the procedure involving the Implanon device.

Reviewer 2 Report

Comments and Suggestions for Authors

This case report is fascinating and provides detailed descriptions of the technique, making it a valuable source of information.
However, the content is highly similar to that of Reference 6. While Reference 6 reports results from 26 cases, this paper presents only one case report, and the authors' claim that it is the “first documented” case is an exaggeration. The authors should clarify the scientific value of their work by explicitly stating that the method described in this paper is based on the method reported in Reference 6 and represents a further development of that method. 
Both the introduction and discussion sections are redundant. Given that this paper's value lies in the detailed description of the technique, these sections should be concise. In particular, when comparing the new method with conventional methods, the authors should use tables to provide an overview of the previous techniques and their results.

Author Response

This case report is fascinating and provides detailed descriptions of the technique, making it a valuable source of information.

Response: We appreciate the reviewer’s positive feedback that our case is “fascinating” and “provides detailed descriptions of the technique, making it a valuable source of information.” We have worked to enhance its value further by incorporating all the requested clarifications and structural improvements.

However, the content is highly similar to that of Reference 6. While Reference 6 reports result from 26 cases, this paper presents only one case report, and the authors' claim that it is the “first documented” case is an exaggeration. The authors should clarify the scientific value of their work by explicitly stating that the method described in this paper is based on the method reported in Reference 6 and represents a further development of that method. 

Response: Thank you for your suggestion. While previous studies have described ultrasound-guided removal of deep implants [6, 8], our technique represents a significant advancement by integrating 5% dextrose in water (D5W) hydrodissection to separate the nerve from the implant, thereby creating a safety zone for ultrasound-guided manipulation of the implant with surgical instruments and preventing further nerve damage. This approach also reduces neurogenic inflammation associated with implant manipulation. Additionally, our technique introduces percutaneous needle stabilization specifically designed for easily migrated or slippery implants. To our knowledge, this is the first documented application of these combined approaches for the removal of contraceptive implants, effectively addressing mechanical impingement, neuroinflammation, and the technical challenges posed by slippery implants, all without the need for open surgery.

  • We have added in the manuscript, lines 666 to 687: “ While previous studies have described ultrasound-guided removal of deep im-plants [6, 8], our technique represents a significant advancement by integrating 5% dex-trose in water (D5W) hydrodissection to separate the nerve from the implant, thereby creating a safety zone for ultrasound-guided manipulation of the implant with surgical instruments and preventing further nerve damage. This approach also reduces neuro-genic inflammation associated with implant manipulation. Additionally, our technique introduces percutaneous needle stabilization specifically designed for easily migrated or slippery implants.

This is, to our knowledge, the first documented case of successful management of an iatrogenically malpositioned implant, addressed with ultrasound-guided hydrodis-section using D5W to mitigate mechanical impingement and neurogenic inflammation. The implant was successfully removed, despite its slippery nature, via ultrasound-guided percutaneous extraction with a single needle stabilization, obviating the need for additional incisions or open surgery.”

Both the introduction and discussion sections are redundant. Given that this paper's value lies in the detailed description of the technique, these sections should be concise.

Response: Thank you for your suggestions. We have condensed the introduction and discussion sections accordingly.

In particular, when comparing the new method with conventional methods, the authors should use tables to provide an overview of the previous techniques and their results.

Response: Thank you for your suggestions. We have added table 1.

Table 1. Comparative Analysis of Deeply-Located Implants Removal Techniques

Characteristic

Open Removal

US-Guided Removal

US-Guided HD + Stabilization + Removal

Target Indications

Excessively difficult implants

Deep implants (not mentioning distance from nearby nerves in literature)

Nerve-adjacent or impinging implants

Operation Time

Long (>2 hours)

Short (<30 minutes)

Moderate (30 to 60 minutes)

Scar Size

Large incision

Small incision

Small incision

Nerve Injury Risk

Possible

Possible

Minimal

Soft Tissue Damage

Extensive

Moderate

Minimal

Migration Risk

Negligible

Possible

Reduced

Conversion to Open

N/A

Possible

Rare

Learning Curve

Low

Moderate

High

Complication Rate

Higher

Lower

Minimal

Table 1: This table summarizes the key differences between traditional open removal, ultrasound-guided removal, and ultrasound-guided removal with hydrodissection (HD) and stabilization techniques.

Round 2

Reviewer 1 Report

Comments and Suggestions for Authors

The required changes have been made, and I have no further comments.

Author Response

Thank you very much for taking the time to review our manuscript. We greatly appreciate it.

Reviewer 2 Report

Comments and Suggestions for Authors

The authors have responded appropriately to the peer review comments and have made substantial improvements to the revised manuscript. The content is potentially useful for the readership.
However, the manuscript still contains numerous instances of overly elaborate expressions that may detract from its scientific clarity. It is recommended that the authors seek the assistance of a professional English editor with extensive experience in academic writing to address these issues and enhance the overall quality of the manuscript.

Comments on the Quality of English Language

The manuscript still contains numerous instances of overly elaborate expressions that may detract from its scientific clarity. It is recommended that the authors seek the assistance of a professional English editor with extensive experience in academic writing to address these issues and enhance the overall quality of the manuscript.

Author Response

Thank you very much for taking the time to review our manuscript. We greatly appreciate it.

Our manuscript, apart from being edited for English proficiency by Editage, has also been edited by the MDPI English Editors